# Structural mechanisms of TRPV6 inhibition by ruthenium red and econazole

Arthur Neuberger [1,2], Kirill D. Nadezhdin [1,2] & Alexander I. Sobolevsky [1✉]

TRPV6 is a calcium-selective ion channel implicated in epithelial $Ca^{2+}$ uptake. TRPV6 inhibitors are needed for the treatment of a broad range of diseases associated with disturbed calcium homeostasis, including cancers. Here we combine cryo-EM, calcium imaging, and mutagenesis to explore molecular bases of human TRPV6 inhibition by the antifungal drug econazole and the universal ion channel blocker ruthenium red (RR). Econazole binds to an allosteric site at the channel's periphery, where it replaces a lipid. In contrast, RR inhibits TRPV6 by binding in the middle of the ion channel's selectivity filter and plugging its pore like a bottle cork. Despite different binding site locations, both inhibitors induce similar conformational changes in the channel resulting in closure of the gate formed by S6 helices bundle crossing. The uncovered molecular mechanisms of TRPV6 inhibition can guide the design of a new generation of clinically useful inhibitors.

[1] Department of Biochemistry and Molecular Biophysics, Columbia University, New York, NY, USA. [2] These authors contributed equally: Arthur Neuberger, Kirill D. Nadezhdin. ✉email: as4005@cumc.columbia.edu

TRPV6 is a representative of the vanilloid subfamily of transient receptor potential (TRP) channels that is considered the principal epithelial calcium channel for $Ca^{2+}$ absorption in intestines and many other organs[1]. TRPV6 is highly selective to $Ca^{2+}$ ($P_{Ca}/P_{Na} > 100$)[2–5] and its constitutive activity is regulated positively by membrane lipids, such as phosphatidylinositol 4,5-bisphosphate ($PIP_2$)[6–8], and negatively by $Mg^{2+}$, which is at least partially responsible for the strong inward rectification displayed by these channels[9,10], and calmodulin (CaM) that causes $Ca^{2+}$-dependent inactivation[11–13] by plugging the channel through a unique cation-π interaction by inserting the side chain of lysine K115 into a tetra-tryptophan cage at the pore's intracellular entrance[14]. Dysregulation of TRPV6 leads to disturbed calcium homeostasis. Thus, knockout TRPV6$^{-/-}$ mice exhibit defective intestinal $Ca^{2+}$ absorption, increased urinary $Ca^{2+}$ excretion, decreased femoral bone mineral density, lower body weight, alopecia, dermatitis, and severely impaired male fertility[15–19], whereas changes in TRPV6 expression have been shown in several mouse models of human diseases[20–24]. Decreased mRNA and protein levels of TRPV6 have been found in placentas of women suffering from pre-eclampsia[25]. Several human TRPV6 gene mutations have been recently linked to transient neonatal hyperparathyroidism, skeletal under-mineralization and dysplasia, and chronic pancreatitis[26–32]. Because of the central role of calcium in cancer development[33], TRPV6 has also been classified as an oncochannel that is involved in both increased cell proliferation and inhibition of apoptosis[34], and overexpressed in various types of human cancer, including breast, prostate, colon, ovarian, thyroid, endometrial cancers, and leukemia[33,35]. TRPV6 inhibitors can therefore be used for the treatment of a broad range of diseases associated with disturbed calcium homeostasis, including TRPV6-rich tumors[36–39].

Several small-molecule inhibitors of TRPV6 have been identified, including 2-aminoethoxydiphenyl borate (2-APB) and TH-1177, their derivatives, highly potent and selective derivatives of (4-phenylcyclohexyl)piperazine (PCHPDs), trinuclear ruthenium amine ruthenium red (RR), and antifungal drugs econazole and miconazole[2,40–47]. Structures of TRPV6 identified a 2-APB-binding site at the cytoplasmic end of the S1–S4 transmembrane helical bundle and suggested an allosteric mechanism of channel closure through modulation of protein–lipid interactions[48]. Several structures of TRPV6 in complex with PCHPDs identified their main binding site at the intracellular entrance to the ion channel pore, which overlaps with the binding site of CaM[49]. Therefore, PCHPDs act as TRPV6 open-channel blockers that convert the channel into a non-conducting state, mimicking inactivation by CaM[14,49].

Here we use cryo-EM, calcium imaging, and mutagenesis to study structural mechanisms of TRPV6 inhibition by econazole and RR. RR, which is used in many biological applications[50–52], inhibits a broad spectrum of ion channels, including ryanodine receptors, mitochondrial calcium uniporters, calcium, $K_{2P}$, TRP, CALHM, and Piezo channels[53,54]. RR appears to act as a channel blocker of TRPV6 but compared with PCHPDs, its binding site is located on the opposite side of the ion channel pore, in the selectivity filter. In contrast, econazole acts as an allosteric inhibitor of TRPV6, similar to 2-APB. However, econazole binds between S1-S4 bundle of one TRPV6 subunit and S5 of the neighboring subunit, at the transmembrane domain (TMD) location that is different from the 2-APB-binding site[48]. Nevertheless, econazole replaces a lipid that is otherwise present at the binding site and induces a similar closure of the ion channel gate as do RR and 2-APB. Our results expand the spectrum of inhibitor binding sites and molecular mechanisms of inhibition that can be explored for the design of new drugs targeting TRPV6.

## Results

**Structure of human TRPV6 in the open state**. To study molecular details of TRPV6 inhibition, we first optimized our protein preparation. As a control, we aimed to determine the open (apo) state structure of human TRPV6 (TRPV6$_{Open}$) at a higher resolution than that of the previously solved structures[49,55]. To avoid possible interference of inhibition with CaM-mediated inactivation, we used the C-terminally truncated functional human TRPV6 construct (hTRPV6-CtD) that lacks the CaM-binding site[49]. Initially, we improved the structural resolution for hTRPV6-CtD purified in glyco-diosgenin (GDN) detergent from 3.26 Å[49] to 2.69 Å (Fig. 1a, Supplementary Fig. 1, Supplementary Table 1). As an alternative to the detergent, we reconstituted hTRPV6-CtD into cNW11 nanodiscs. This further improved the resolution to 2.54 Å (Supplementary Fig. 1, Supplementary Table 1), resulting in a structure that is essentially identical to the one in detergent (Supplementary Fig. 2), although with slightly fewer and less well resolved annular lipids. These two hTRPV6-CtD structures were also very similar to TRPV6 structures in amphipols and MSP2N2 nanodiscs (Supplementary Fig. 2), suggesting that TRPV6 structure determination does not depend on the type of membrane mimic or the presence or absence of the C-terminus and yields similar open-state conformations, albeit with a slightly better resolution in cNW11 nanodiscs.

The new open-state structures have the same architecture as previously published TRPV6 structures (Fig. 1b, Supplementary Fig. 2)[14,48,55–57]. The channel is assembled of four subunits and contains two main components: a TMD with a central ion channel pore and an intracellular skirt that is mostly built of ankyrin repeat domains connected to each other by three-stranded β-sheets, N-terminal helices, and C-terminal hooks. Amphipathic TRP helices run nearly parallel to the membrane and interact with both the TMD and the skirt. The TMD is composed of six transmembrane helices S1–S6 and a re-entrant pore loop (P-loop) between S5 and S6. A bundle of the first four transmembrane helices represents the S1–S4 domain. A domain homologous to the S1–S4 domain acts as a voltage sensor in voltage-gated ion channels[58]. The pore domain of each subunit includes S5, P-loop, and S6, and is packed against the S1–S4 domain of the neighboring subunit in a domain-swapped arrangement[55,57].

**Structures of TRPV6 in complex with RR and econazole**. As cNW11 nanodiscs yielded the open-state structure at better resolution than in detergent, amphipols, or MSP2N2 nanodiscs, especially for the protein part, we used cNW11 nanodiscs to determine TRPV6 structures in complex with the inhibitors. Indeed, the resolution of structures obtained in the presence of RR (TRPV6$_{RR}$) and econazole (TRPV6$_{Eco}$) was about the same as for the open-state structure in cNW11 nanodiscs (Fig. 2, Supplementary Fig. 1, Supplementary Table 1). Compared with the open-state structure, the structure solved in the presence of RR showed a distinct non-protein density in the middle of the ion channel pore that perfectly matched the shape of RR (Fig. 2a, c). On the contrary, the structure of TRPV6 determined in the presence of econazole did not show densities in the ion channel pore that would resemble a molecule of econazole. Instead, it had four densities (one per subunit) in the TMD region peripheral to the ion channel pore with the shape that closely matched an econazole molecule (Fig. 2b, d). The location of these densities coincides with the location of lipid in other structures (Supplementary Fig. 3).

**Binding site of RR in TRPV6 channel pore**. Different parts of channel pores serve as binding sites for ion channel blockers. In

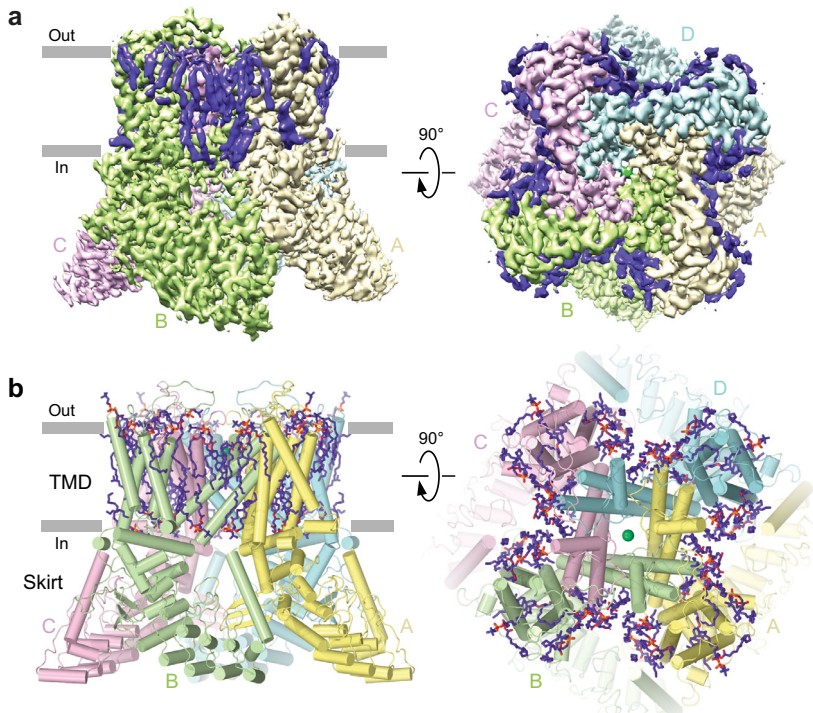

**Fig. 1 Structure of TRPV6 in the open (apo) state. a** Cryo-EM density for TRPV6$_{Open}$, with individual subunits shown in yellow, green, pink, and blue, and lipids in purple. **b** Model of TRPV6$_{Open}$, with lipids shown as sticks and calcium ion as a green sphere.

TRPV6, two types of ion channel blockers have been characterized structurally, gadolinium[56,57] and PCHPDs[49]. Although PCHPDs bind at the intracellular entrance to the ion channel pore, the gadolinium site is located at the extracellular entrance and formed by aspartates D542. The same site is shared by permeable ions[56,57,59] and calcium binds at this site in the open state (Fig. 3a, d). RR binding occurs close to the extracellular pore entrance and its site partially overlaps with the binding site for ions (Fig. 3b, e). High-resolution reconstruction of TRPV6$_{RR}$ allows precise placement of the inhibitor inside the ion channel pore (Fig. 3b). On the extracellular side, the RR-binding site is flanked by aspartates D542 with a spherical density on the top that might represent a calcium ion typically bound at D542 in the absence of RR. The inhibitor occupies the entire selectivity filter of TRPV6, which is lined by carboxyl groups of D542, backbone carbonyl oxygens of I541, I540, and T539 as well as the hydroxyl group of T539, and spans the upper half of the pore's central cavity. The surface of the selectivity filter is highly electronegative (Fig. 3e), creating a favorable environment for the positively charged RR that carries a 6$^+$ total charge. Similar to TRPV6$_{Open}$ structure (Fig. 3a, d), the structure of TRPV6$_{Eco}$ shows no RR-like density in the selectivity filter but a strong density at the D542 site, presumably representing Ca$^{2+}$ (Fig. 3c, f). There is also a noticeable density in the central cavity of TRPV6$_{Eco}$ pore. The shape of this density did not match the shape of econazole, even after multiple rounds of focused classification and symmetry relaxation. Nevertheless, as TRPV6$_{Open}$ and TRPV6$_{RR}$ structures do not contain any density in the central cavity above the noise level, there is a possibility that the observed density in the TRPV6$_{Eco}$ structure represents a low-affinity or non-specific binding of the inhibitor in random orientations that cannot be revealed by cryo-EM-processing tools.

**Allosteric binding site of econazole.** The econazole-binding site is located at the interface between S5 of one subunit and S4 of the neighboring subunit (Fig. 4a, c). Econazole is sandwiched between W495 of S5 and F472 of S4 and is also surrounded by hydrophobic side chains of L496 and V499 of S5 as well as M466, A469, M474, and L475 of S4. It is not surprising that the hydrophobic character of the econazole-binding site creates a favorable environment for lipids, which bind at this site in the absence of econazole (Supplementary Fig. 3; Fig. 4b, d).

We probed the econazole-binding site by mutating residues that contribute to this site and compared the concentration dependence of econazole inhibition of the corresponding mutants with wild-type channels (Fig. 4e). Econazole inhibited Ca$^{2+}$ uptake through wild-type TRPV6 with $IC_{50}$ of $4.4 \pm 0.3\ \mu M$ ($n = 9$). Substantial weakening of inhibition observed for L475A ($IC_{50} = 151 \pm 20\ \mu M$, $n = 3$), F472A ($3.14 \pm 0.24\ mM$, $n = 6$) and W495A ($IC_{50} = 3.43 \pm 0.52\ mM$, $n = 6$) mutants strongly supported the identified binding site of econazole. Mutation T539A in the ion channel pore, which is distal to the econazole-binding site, did not change the efficacy of calcium uptake inhibition by econazole ($IC_{50} = 3.1 \pm 0.2\ \mu M$, $n = 3$), confirming that the effects of L475A, F472A, and W495A mutations were specific. The strong effects of W495A and F472A mutations on econazole inhibition also indicate that the binding site at the S4–S5 interface is the main inhibitory site of econazole. Therefore, if density observed in the pore's central cavity of TRPV6$_{Eco}$ (Fig. 3c) represents econazole molecule(s), it can only justify non-specific binding or low-affinity inhibition.

**Effects of RR and econazole on TRPV6 gating.** To explore the possible effects of RR and econazole on TRPV6 gating, we compared the ion channel pore in TRPV6$_{Open}$ with the pore in TRPV6$_{RR}$ and TRPV6$_{Eco}$ (Fig. 5). Consistent with previous findings, the pore of the open-state TRPV6 was in the conducting state (Fig. 5a, d). Typical for the open conformation of TRPV6, S6 was not entirely α-helical and contained a π-bulge resulting from an α-to-π transition in its middle (Figs. 5a, 3a). In contrast, the pores in TRPV6$_{RR}$ and TRPV6$_{Eco}$ were apparently closed due to a hydrophobic seal imposed by the gate residues L574 and M578

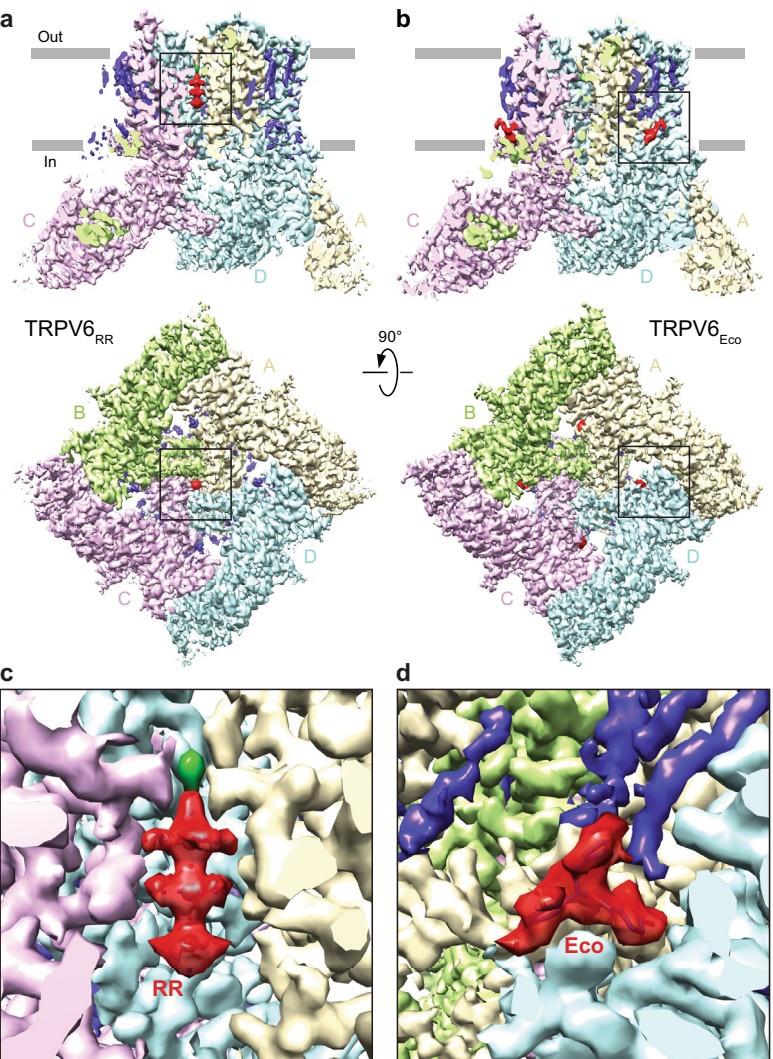

**Fig. 2 Structures of TRPV6 in complex with ruthenium red and econazole. a, b** Cryo-EM structures of TRPV6$_{RR}$ (**a**) and TRPV6$_{Eco}$ (**b**) viewed parallel to the membrane (upper row) or intracellularly (bottom row), with each of the four subunits colored differently, lipids in purple and inhibitors in red. **c, d** Close-up views of cryo-EM density for RR (**c**) and econazole (**d**).

(Figs. 5b–d, 3e, f). Correspondingly, S6 in TRPV6$_{RR}$ and TRPV6$_{Eco}$ was entirely α-helical, typical for the closed conformation of TRPV6. The open-to-closed state transformation of the lower pore in TRPV6$_{Open}$ compared with TRPV6$_{RR}$ and TRPV6$_{Eco}$ included a ~100° rotation and bending away from the pore axis of the intracellular part of S6, as it was described previously[48,55]. This conformational change in S6 started below the gating hinge alanine A566 and did not cause alteration in the upper pore of TRPV6, leaving the selectivity filter apparently intact (Figs. 3 and 5).

## Discussion

By studying structural bases of TRPV6 inhibition by RR and econazole we identified two novel TRPV6 inhibitory binding sites: in the ion channel selectivity filter (Fig. 3) and at the interface between transmembrane helices S4 and S5 (Fig. 4). When added to the previously discovered inhibitory sites of gadolinium formed by the side chains of aspartates D542 at the pore extracellular entrance[56,57], 2-APB in the pocket formed by the cytoplasmic half of the S1–S4 transmembrane helix bundle[48], and PCHPDs at the pore intracellular entrance[49], they increase the number of target sites for potential therapeutic interventions

as well as synergistic treatment approaches. The number of distinct ligand binding sites in the vanilloid subfamily of TRP channels (TRPVs) therefore reaches twelve (Supplementary Fig. 4). Given the similarity in amino-acid sequences (Supplementary Fig. 5) and structural architecture of TRPVs, there is a good chance that all these sites can be targeted in every member of the subfamily. Some of these sites, like the vanilloid binding site initially identified in TRPV1[60], are already found to bind ligands in different TRPVs[61,62]. Other sites might only bind ligands in certain TRPVs and their specificity will require additional studies.

Previously, binding sites of RR were identified structurally in CALHM channels[54] and K$_{2P}$ potassium channels[53]. In undecameric CALHM channels, RR binds near the pore-lining helix S1 and was proposed to stabilize this helix in a lifted conformation, giving rise to channel inhibition[54]. In K$_{2P}$ channels, RR binds under the CAP domain archway above the channel pore and was proposed to act according to a "finger in the dam" mechanism[53]. The two proposed mechanisms are fundamentally different from the mechanism of TRPV6 inhibition, where RR acts like a cork in the bottle by plugging the pore in the narrow region of the selectivity filter (Fig. 6). This region of the pore is likely optimal

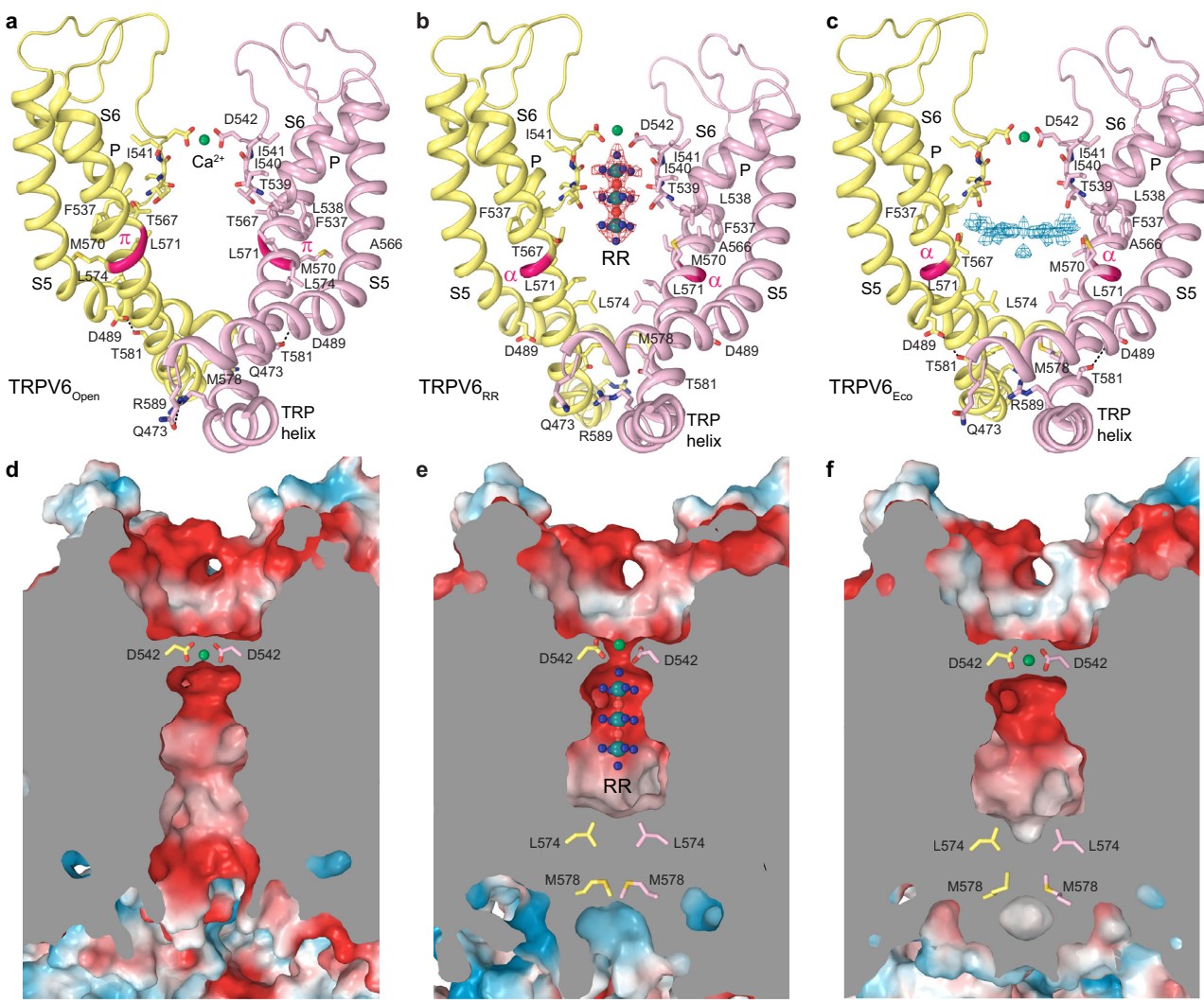

**Fig. 3 Binding site of the ion channel blocker RR. a–c** Structure of the ion channel pore in TRPV6_Open (**a**), TRPV6_RR (**b**), and TRPV6_Eco (**c**) viewed parallel to the membrane. Only two of four subunits are shown, with the front and back subunits omitted for clarity. Also shown is the molecule of RR and residues contributing to RR binding, calcium selectivity, gating, or being involved in state-specific interactions. Red and blue mesh shows density for RR and non-protein density in the TRPV6_Eco pore, respectively. Interactions between Q473 and R589 as well as D489 and T581 are indicated by dashed lines. The region that undergoes α-to-π transition in S6 is highlighted in bright pink. Calcium ions are shown as green spheres. **d–f** Coronal sections of the pore in TRPV6_Open (**d**), TRPV6_RR (**e**), and TRPV6_Eco (**f**), with the surface colored by electrostatic potential.

for RR block owing to its unique geometry as well as the overall negative surface charge that complements the positive charge of the blocker (Fig. 3).

Previous structural determination of econazole-binding sites in ion channels is limited to TRPV5 channels, which are highly homologous to TRPV6 (Supplementary Fig. 5). Based on the TRPV5 structure determined in the presence of econazole[62], econazole was proposed to bind to the vanilloid site, which in TRPV1 accommodates vanilloid ligands like agonists resiniferatoxin and capsaicin, competitive antagonist capsazepine, or phosphatidylinositol (PI) lipid[60,63,64]. The vanilloid binding site is different from the econazole-binding site in TRPV6 identified in the present study. All residues coordinating econazole at its binding site in TRPV6 are conserved between TRPV5 and TRPV6, except A469, which is a threonine in TRPV5 (Supplementary Fig. 5). It is therefore unclear why econazole binding in these two highly homologous channels would be different. In all high-resolution structures of TRPV6, the vanilloid site is clearly occupied by cholesteryl hemisuccinate (CHS), which is routinely used during TRPV6 purification[49]

(Supplementary Fig. 3), and likely accommodates cholesterol in cellular conditions. It is therefore possible that in a relatively low, 4.8-Å resolution reconstruction of TRPV5, the blob of density assigned to the molecule of econazole[62] represents a lipid. Higher-resolution structural information about TRPV5 inhibition by econazole would be helpful to better understand the difference from TRPV6. Of other inhibitory binding sites in TRP channels, the location of the econazole-binding site in TRPV6 (Fig. 4) has the highest similarity with the location of a piperlongumine-binding site in TRPV2[65]. Compared with the econazole-binding site in TRPV6, the piperlongumine-binding site in TRPV2 is wedged deeper between S4 and S5 and makes contact with S6.

Compared with RR, which directly occludes the channel pore and prevents ionic conductance, econazole acts as an allosteric inhibitor. Econazole binds to the site, which is occupied by a lipid in the apo-state structure (Fig. 4). Because of the smaller size of econazole compared to the lipid, econazole binding creates a void that allows Q473 and M474 side chains to move closer to F472 and W495 and correspondingly away from R589 (Fig. 4f).

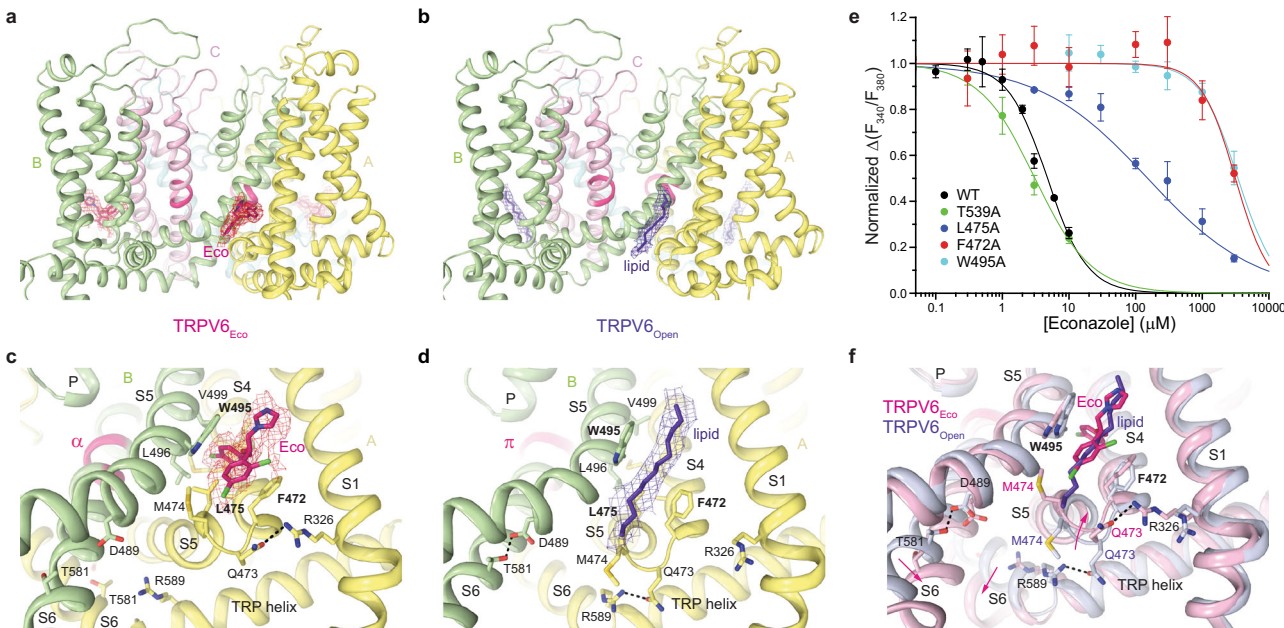

**Fig. 4 Allosteric binding site of econazole. a, b** TMD in TRPV6_Eco (**a**) and TRPV6_Open (**b**) viewed parallel to the membrane. **c, d** Close-up view of the econazole-binding site in TRPV6_Eco (**c**) and TRPV6_Open (**d**). The molecule of econazole, residues contributing to its binding, and the lipid occupying the peripheral site in the open-state structure are shown as sticks. Dashed lines represent hydrogen bonds or salt bridges between residues. **e** Dose–response curves for econazole inhibition of $Ca^{2+}$ uptake calculated for wild-type (WT) and mutant TRPV6 channels. The changes in the fluorescence intensity ratio at 340 and 380 nm ($F_{340}/F_{380}$) evoked by the addition of 5 mM $Ca^{2+}$ after pre-incubation with various concentrations of econazole were normalized to the maximal change in $F_{340}/F_{380}$ after addition of 5 mM $Ca^{2+}$ in the absence of econazole. Straight lines through the data points are fits with the logistic equation, with the mean ± SEM values of the half-maximum inhibitory concentration ($IC_{50}$), 4.39 ± 0.31 μM for WT ($n = 9$), 3.09 ± 0.22 μM for T539A ($n = 3$), 151 ± 20 μM for L475A ($n = 3$), 3.14 ± 0.24 mM for F472A ($n = 6$) and 3.43 ± 0.52 mM for W495A ($n = 6$). Source data are provided as a Source Data file. **f** Superposition of the econazole-binding site region in TRPV6_Eco (pink) and TRPV6_Open (purple). Pink arrows illustrate the movement of the S4–S5 linker and S6 upon econazole binding.

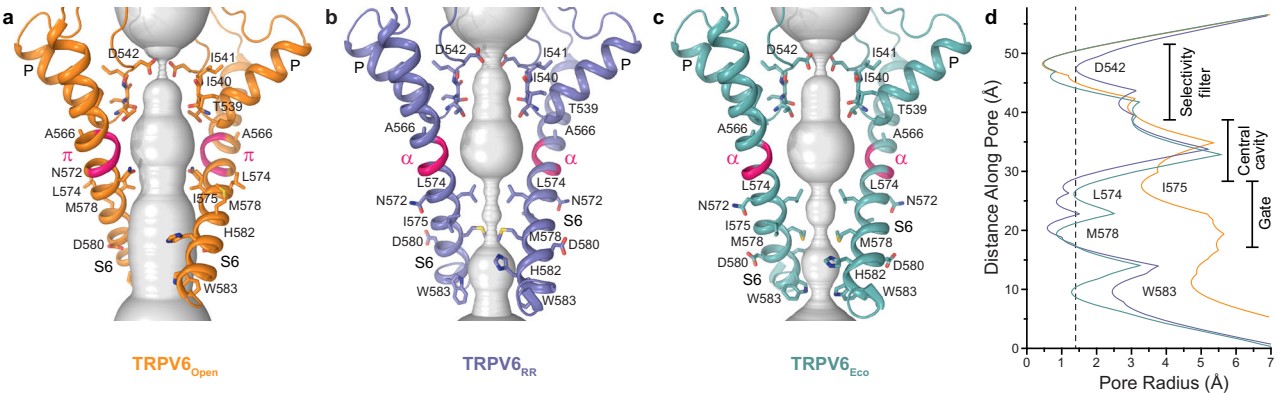

**Fig. 5 Ion channel pore. a–c** Pore-forming domains in the open (**a**), RR-bound (**b**), and econazole-bound (**c**) states with the residues lining the pore shown as sticks. Only two of four subunits are shown, with the front and back subunits omitted for clarity. The pore profiles are shown as space-filling models (light gray). The region that undergoes α-to-π transition in S6 is highlighted in pink. **d** Pore radius for TRPV6 structures calculated using HOLE. The vertical dashed line denotes the radius of a water molecule, 1.4 Å.

Separation of Q473 and R589 results in breakage of the salt bridge connecting these residues in the open state[55]. The loss of this salt bridge, which energetically compensates for the unfavorable α-to-π transition in S6, reverses the transition. Concurrently, the lower portions of S6 helices rotate by ~100°, leading to the separation of D489 and T581 and the loss of the open-state-stabilizing hydrogen bond between them[55]. The rotated lower portions of S6 expose the side chains of L574 and M578 towards the pore center and hydrophobically seal the pore to prevent ion conductance, thus converting the channel into the closed, non-conducting state (Figs. 3, 5, and 6).

What remains a puzzle is why RR, which binds at the selectivity filter and physically occludes this region of the pore for ion conductance, also causes the gating transition in the lower pore similar to econazole. The selectivity filter conformations in TRPV6_RR, TRPV6_Eco, and TRPV6_Open are nearly identical, as well as of the P-helix that directly contacts S6. We hypothesize that the positive charge of RR creates an electric field inside the pore's central cavity that interacts with the electric dipole of the S6 helix (Fig. 6a). This interaction causes repulsion of the lower portion of S6 away from RR, which results in partial S6 rotation and helps S6 to become completely α-helical, with side chains of L574 and

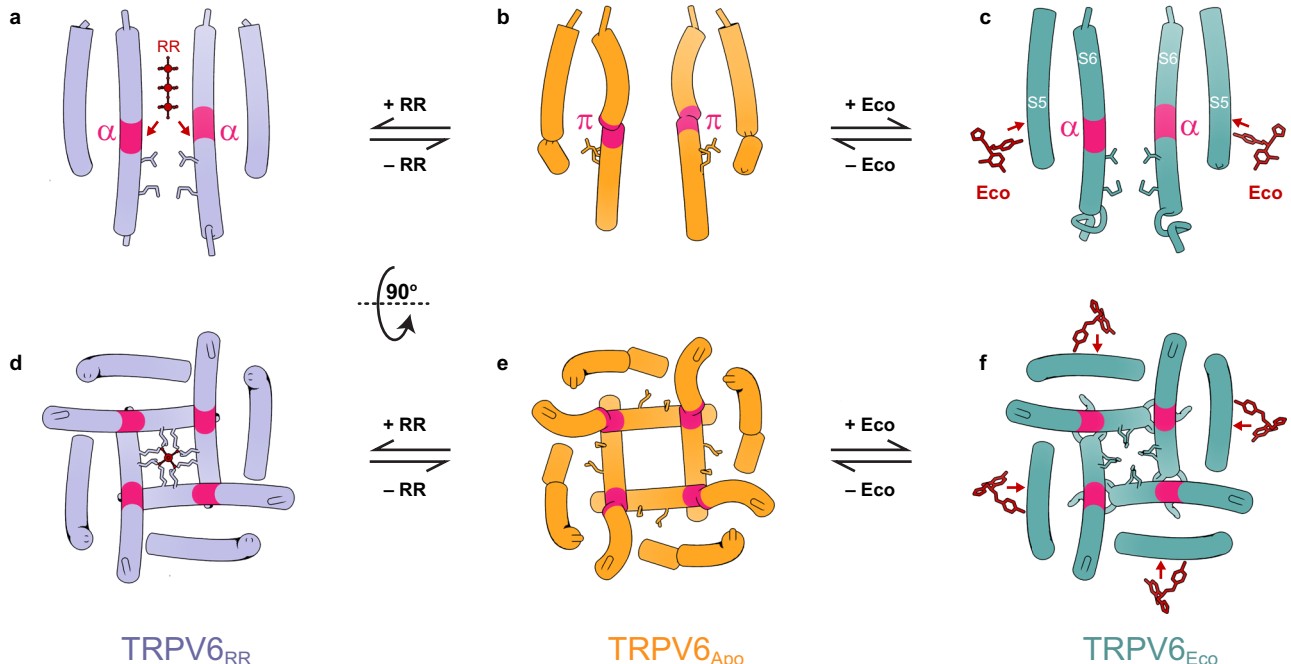

**Fig. 6 Mechanisms of inhibition.** Side (**a–c**) and bottom (**d–f**) views of the ion channel pore in TRPV6$_{RR}$ (**a**, **d**), TRPV6$_{Open}$ (**b**, **e**), and TRPV6$_{Eco}$ (**c**, **f**). Only two of four subunits are shown in **a–c**, with the front and back subunits omitted for clarity. Residues that form the gate in the closed channel and molecules of RR and econazole (Eco) are shown as sticks. Arrows illustrate the directionality of force exerted by RR and econazole on S6 and S5, respectively, to close the ion channel.

M578 sealing the pore (Fig. 6a, d). Interestingly, all currently known TRPV6 inhibitors either directly block the ion channel pore (Ga$^{3+}$, calmodulin, PCHPDs, RR) or produce an allosteric effect by displacing lipids (2-APB, econazole). More work and high-resolution structures will be necessary to reveal novel druggable sites on the surface of TRPV6 and better understand the mechanisms and energetics of TRPV6 inhibition.

With RR and econazole joining PCHPDs and 2-APB in the list of TRPV6 inhibitors with structurally identified binding sites, there is a stronger foundation for the structure-based design and development of new therapeutic agents potentially capable of targeting a broad spectrum of TRPV6-linked diseases, including various types of human cancer. Importantly, since all of these compounds occupy distinct binding sites, they can likely bind simultaneously, resulting in synergistic inhibition of TRPV6. Thus, co-administration of molecules representing these different classes of inhibitors might be a future strategy for a more efficient, disease-dependent, and fine-tuned personalized therapy approach. The close homology and sequence similarity between TRPV6 and TRPV5 (Supplementary Fig. 5) suggest that these approaches can also be extended to TRPV5-linked diseases, including hypercalciuria, nephrolithiasis, and bone disorders[66,67].

## Methods

**Constructs**. C-terminally truncated human TRPV6 (hTRPV6-CtD, residues 1–666 of wild-type channel)[49] used for cryo-EM was cloned into a pEG BacMam vector[68] with a C-terminal thrombin cleavage site followed by a streptavidin affinity tag (WSHPQFEK). For Fura-2 AM measurements, point mutations in wild-type human TRPV6 were introduced using standard molecular biology techniques (see the list of primers in Supplementary Table 2).

**Expression and purification**. hTRPV6-CtD was expressed and purified based on our previously developed protocol[49]. Bacmids and baculoviruses were produced using standard procedures[68]. Baculovirus was made in Sf9 cells for ~72 hours (Thermo Fisher Scientific, mycoplasma test negative, GIBCO #12659017) and was added to suspension-adapted HEK 293 cells lacking N-acetyl-glucosaminyltransferase I (GnTI⁻, mycoplasma test negative, ATCC #CRL-3022) that were maintained in Freestyle 293 media (Gibco-Life Technologies #12338-018)

supplemented with 2% FBS at 37 °C and 5% CO$_2$. Twenty-four hours after transduction, 10 mM sodium butyrate was added to enhance protein expression, and the temperature was reduced to 30 °C. Seventy-two hours after transduction, cells were harvested by centrifugation at 5471 × g for 15 min using a Sorvall Evolution RC centrifuge (Thermo Fisher Scientific), washed in phosphate-buffered saline pH 8.0, and pelleted by centrifugation at 3202 × g for 10 min using an Eppendorf 5810 centrifuge. The cell pellet was resuspended in ice-cold lysis buffer containing 20 mM Tris-Cl pH 8.0, 150 mM NaCl, 0.8 μM aprotinin, 4.3 μM leupeptin, 2 μM pepstatin A, 1 mM phenylmethylsulfonyl fluoride, and 1 mM β-mercaptoethanol (βME). Cells were subsequently lysed using a Misonix Sonicator with a preset program (six cycles of 15 s "on" at the amplitude of eight followed by 15 s "off"; this program was repeated three times for optimal cell lysis) under constant stirring on ice. Unbroken cells and cell debris were pelleted in an Eppendorf 5810 centrifuge at 3202 × g and 4 °C for 10 min. The supernatant was subjected to ultracentrifugation in a Beckman Coulter ultracentrifuge using a Beckman Coulter Type 45Ti rotor at 186,000 g and 4 °C for 1 hour to pellet the membranes. The membrane pellet was mechanically homogenized and solubilized in the lysis buffer supplemented with 1% (w/v) n-dodecyl β-ᴅ-maltoside and 0.1% (w/v) CHS under stirring at 4 °C for 2 hours. Insoluble material was removed by ultracentrifugation for 40 min in a Beckman Coulter Type 45Ti rotor at 186,000 × g and the supernatant was added to strep resin and rotated for 14–16 hours at 4 °C. The resin was washed with 10 column volumes of wash buffer containing 20 mM Tris-HCl pH 8.0, 150 mM NaCl, 1 mM βME, 0.01% (w/v) GDN, and 0.001% (w/v) CHS, and the protein was eluted with the same buffer supplemented with 2.5 mM D-desthiobiotin. The eluted protein was concentrated using a 100 kDa NMWL centrifugal filter (MilliporeSigma™ Amicon™) to 0.5 ml and then centrifuged in a Sorvall MTX 150 Micro-Ultracentrifuge (Thermo Fisher Scientific) using an S100AT4 rotor for 30 min at 66,000 × g and 4 °C before injection into a size-exclusion chromatography (SEC) column. All constructs were further purified using a Superose™ 6 10/300 GL SEC column attached to an AKTA FPLC (GE Healthcare) and equilibrated in 150 mM NaCl, 20 mM Tris-HCl pH 8.0, 1 mM βME, 0.01% GDN, and 0.001% CHS. The tetrameric peak fractions were pooled and concentrated using 100 kDa NMWL centrifugal filter to 2–4 mg/ml.

Circularized cNW11 nanodiscs were prepared according to the standard protocol[69] and stored (~2–3 mg/ml) before usage at –80 °C in 20 mM Tris (pH 8.0) and 150 mM NaCl. For nanodisc reconstitution, the purified protein was mixed with circularized cNW11 nanodiscs and soybean lipids (Soy polar extract, Avanti Polar Lipids, USA) at a molar ratio of 1:3:166 (TRPV6 monomer:cNW11:lipid). The lipids were dissolved to a concentration of 100 mg/ml in 150 mM NaCl and 20 mM Tris (pH 8), and subjected to 5–10 cycles of freezing in liquid nitrogen and thawing in a water bath sonicator. For TRPV6$_{RR}$, 1 mM RR was added to the TRPV6 monomer:cNW11:lipid mixture. The nanodisc mixture (500 μl) was rocked at room temperature for 1 h. Subsequently, 20 mg of Bio-beads SM2 (Bio-Rad), pre-wet in a buffer containing 20 mM Tris pH 8, 150 mM NaCl and 1 mM βME,

were added to the nanodisc mixture, which was then rotated for one hour at 4 °C. Additional 20 mg of Bio-beads SM2 were added and the resulting mixture was rotated at 4 °C for another ~14–20 hours. Bio-beads SM2 were then removed by pipetting and nanodisc-reconstituted hTRPV6-CtD was purified from empty nanodiscs by SEC using Superose™ 6 10/300 GL SEC column equilibrated in 150 mM NaCl, 20 mM Tris (pH 8.0), and 1 mM βME. Fractions of nanodisc-reconstituted hTRPV6-CtD were pooled and concentrated using 100 kDa NMWL centrifugal filter to 0.6–3.0 mg/ml.

RR was dissolved in water to a final concentration of 50 mM. Econazole was dissolved in dimethyl sulfoxide to a final concentration of 100 mM. These stocks were added to the cNW11-reconstituted protein at the final concentrations of 0.5 mM econazole or 1 mM RR, and the resulting samples were incubated for two hours at room temperature before grid preparation.

**Cryo-EM sample preparation and data collection**. UltrAuFoil R 1.2/1.3, Au 300 grids were used for plunge-freezing. Prior to sample application, grids were plasma treated in a PELCO easiGlow glow discharge cleaning system (0.39 mBar, 15 mA, "glow" 25 s, "hold" 10 s). A Mark IV Vitrobot (Thermo Fisher Scientific) set to 100% humidity at 4 °C was used to plunge-freeze the grids in liquid ethane after applying 3 μl of protein sample to their gold-coated side using a blot time of 5 s, a blot force of 5, and a wait time of 15 s. To achieve optimal particle distribution for TRPV6$_{RR}$, 3 μl of the protein sample was applied twice. The grids were stored in liquid nitrogen before imaging.

Images of frozen-hydrated particles of TRPV6$_{Open}$ in GDN and CHS were collected on a Titan Krios transmission electron microscope (TEM) (Thermo Fisher Scientific) operating at 300 kV and equipped with a post-column GIF Quantum energy filter and a Gatan K3 Summit direct electron detection (DED) camera (Gatan, Pleasanton, CA, USA) using EPU software (Thermo Fisher). 7904 micrographs were collected in counting mode with an image pixel size of 0.855 Å across a defocus range of −1.0 to −2.0 μm. The total dose of ~58 e$^-$ Å$^{-2}$ was attained by using a dose rate of ~14.1 e$^-$ pixel$^{-1}$ s$^{-1}$ across 50 frames for a 3.0-s total exposure time.

Images of frozen-hydrated particles of cNW11-reconstituted TRPV6$_{Open}$ were collected on a Titan Krios TEM operating at 300 kV with a post-column GIF Quantum energy filter of 20 eV and a Gatan K3 Summit DED camera using SerialEM[70]. 5706 micrographs were collected in super-resolution mode with an image pixel size of 0.413 Å across a defocus range of −0.8 to −2.0 μm. The total dose of ~60 e$^-$ Å$^{-2}$ was attained by using a dose rate of ~16 e$^-$ pixel$^{-1}$ s$^{-1}$ across 50 frames for a 2.0-s total exposure time.

Images of frozen-hydrated particles of cNW11-reconstituted TRPV6$_{RR}$ were collected on a Titan Krios TEM operating at 300 kV with a post-column GIF Quantum energy filter of 20 eV and a Gatan K3 Summit DED camera using SerialEM. 4431 micrographs were collected in super-resolution mode with an image pixel size of 0.4 Å across a defocus range of −0.5 to −2.0 μm. The total dose of ~58 e$^-$ Å$^{-2}$ was attained by using a dose rate of ~16 e$^-$ pixel$^{-1}$ s$^{-1}$ across 50 frames for a 2.0-s total exposure time.

Images of frozen-hydrated particles of cNW11-reconstituted TRPV6$_{Eco}$ were collected on a Titan Krios TEM operating at 300 kV with a post-column GIF Quantum energy filter of 20 eV and a Gatan K3 Summit DED camera using SerialEM. 3950 micrographs were collected in counting mode with an image pixel size of 0.86 Å across a defocus range of −0.8 to −2.0 μm. The total dose of ~60 e$^-$ Å$^{-2}$ was attained by using a dose rate of ~16 e$^-$ pixel$^{-1}$ s$^{-1}$ across 50 frames for a 2.2-s total exposure time.

**Image processing and 3D reconstruction**. All data sets were processed in RELION[71] and cryoSPARC[72] (see Supplementary Table 1 for details). Movies were motion-corrected with MotionCor2[73] algorithm implemented in RELION. Contrast transfer function (CTF) estimation was performed on non-dose-weighted micrographs using Gctf[74]. Subsequent data processing was done on dose-weighted micrographs. Following CTF estimation, micrographs were manually inspected and those with outliers in defocus values, ice thickness, and astigmatism as well as micrographs with lower predicted CTF-correlated resolution (>6 Å) were excluded from the rest of the processing pipeline (individually assessed for each parameter relative to overall distribution; no set threshold). Particles were picked using a Laplacian-of-Gaussian approach or with the help of unpublished TRPV6 cryo-EM density templates in RELION and 2D-classified in RELION or cryoSPARC. A selection of 2D classes was used to generate templates to be used for template-based picking. Picked particles were further 2D- and 3D-classified in iterative classification and selection rounds. Selected 3D classes were used for 3D template generation and refinement. The reported resolutions of the final maps were estimated using the gold-standard Fourier shell correlation. The local resolution predictions were made in RELION, with the resolution range estimated using the FSC = 0.143 criterion[71]. EM density visualization was done in UCSF Chimera[75] and UCSF ChimeraX[76].

As a representative of the image processing workflow, data for TRPV6$_{RR}$ were processed as follows. Initially, 997,105 particles were automatically picked using Laplacian-of-Gaussian approach in RELION. These particles were subjected to several rounds of 2D classification and 2D-class selection to generate 2D templates for template-based picking in RELION. After template-based picking, 584,265 particle images (2x-binned, 0.8 Å/pix) were imported into cryoSPARC for several

iterative rounds of 2D classification, 2D-class selection, and 3D classification (heterogeneous refinement) to eliminate distorted particles. Particle images from the final 3D classification (329,613 particles) were imported back to RELION and subjected to 3D auto-refinement with C1-symmetry without masking, and with the previously published low-pass filtered to 40 Å map of hTRPV6-CtD in the apo-state (EMD-22622) as a reference. The resulting 3.53-Å reconstruction was used as a template for further processing. The refined particles were iteratively CTF-corrected for anisotropic magnification, beamtilt, and defocus, subjected to Bayesian polishing, and refined to 2.97-Å resolution with C1-symmetry and a soft mask. Particles from the last step were subjected to 3D classification into 10 classes using a soft mask and a 2.97-Å map as a reference. Particles from the two highest resolution classes (154,831) were refined to 2.43-Å resolution with C4 symmetry and a soft mask. The map from the final 3D reconstruction was automatically sharpened using RELION postprocessing and further used for model building.

**Model building**. To build TRPV6 models in Coot[77], we used the previously published crystal or cryo-EM structures of TRPV6 as guides[49]. The models were tested for overfitting by shifting their coordinates by 0.5 Å (using shake) in Phenix[78], refining each shaken model against a corresponding unfiltered half map, and generating densities from the resulting models in Chimera. Structures were visualized and figures were prepared in UCSF Chimera, UCSF ChimeraX, and Pymol[79]. The pore radius was calculated using HOLE[80].

**Fura-2 AM measurements**. Wild-type or mutant human TRPV6 was expressed in HEK cells as described above. 48–72 hours after transduction, cells were harvested by centrifugation at 550 × g for 5 mins. The cells were resuspended in prewarmed HEPES-buffered saline (HBS: 118 mM NaCl, 4.8 mM KCl, 1 mM MgCl$_2$, 5 mM D-glucose, 10 mM HEPES pH 7.4) containing 5 μg/ml of Fura-2 AM (Life Technologies) and incubated at 37 °C for 45 minutes. The loaded cells were then centrifuged for 5 mins at 550 × g, resuspended again in prewarmed HBS, and incubated at 37 °C for 20–30 mins in the dark. The cells were subsequently pelleted and washed twice, then resuspended in HBS for experiments. The cells were kept on ice in the dark for a maximum of ~2 hours before fluorescence measurements, which were conducted using spectrofluorometer QuantaMaster™ 40 (Photon Technology International) at room temperature in a quartz cuvette under constant stirring. Intracellular Ca$^{2+}$ was measured by taking the ratio of fluorescence measurements at two excitation wavelengths (340 and 380 nm) and one emission wavelength (510 nm). The excitation wavelength was switched at 200-ms intervals.

**Reporting summary**. Further information on research design is available in the Nature Research Reporting Summary linked to this article.

## Data availability

All data needed to evaluate the conclusions in the paper are present in the paper and/or the Supplementary Information. Cryo-EM density maps have been deposited to the Electron Microscopy Data Bank (EMDB) under the accession codes EMD-24890 [https://www.emdataresource.org/EMD-24890] (TRPV6$_{open}$, GDN/CHS), EMD-24891 [https://www.emdataresource.org/EMD-24891] (TRPV6$_{open}$, cNW11), EMD-24892 [https://www.emdataresource.org/EMD-24892] (TRPV6$_{RR}$, cNW11), and EMD-24893 [https://www.emdataresource.org/EMD-24893] (TRPV6$_{Eco}$, cNW11) (see Supplementary Table 1). The corresponding model coordinates have been deposited to the PDB under accession numbers 7S88 [https://doi.org/10.2210/pdb7S88/pdb] (TRPV6$_{open}$, GDN/CHS), 7S89 [https://doi.org/10.2210/pdb7S89/pdb] (TRPV6$_{open}$, cNW11), 7S8B [https://doi.org/10.2210/pdb7S8B/pdb] (TRPV6$_{RR}$, cNW11), and 7S8C [https://doi.org/10.2210/pdb7S8C/pdb] (TRPV6$_{Eco}$, cNW11) (see Supplementary Table 1). All other data are available from the corresponding author upon request. Source data are provided with this paper.

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

## Acknowledgements

We thank Harry Scott, Omar Davulcu (Pacific Northwest Center for Cryo-EM), Corey Hecksel (Stanford-SLAC), Ulrich Baxa, and Thomas Edwards (National Cancer Institute, The Frederick National Laboratory) for help with microscope operation and data collection. This research was, in part, supported by the National Cancer Institute's National Cryo-EM Facility at the Frederick National Laboratory for Cancer Research under contract HSSN261200800001E. Some of this work was performed at the Stanford-SLAC Cryo-EM Center (S2C2) supported by the NIH Common Fund Transformative High-Resolution Cryo-Electron Microscopy program (U24 GM129541). A portion of this research was supported by NIH grant U24GM129547 and performed at the PNCC at OHSU and accessed through EMSL (grid.436923.9), a DOE Office of Science User Facility sponsored by the Office of Biological and Environmental Research. A.N. is a Walter Benjamin Fellow funded by the Deutsche Forschungsgemeinschaft (DFG, German Research Foundation)–464295817. A.I.S. was supported by the NIH (R01 CA206573, R01 NS083660, R01 NS107253) and NSF (1818086).

## Author contributions

A.N. and K.D.N. made constructs, performed Fura-2 AM measurements, prepared protein samples, and carried out cryo-EM data collection and processing. A.N., K.D.N., and A.I.S. analyzed structural data. A.N., K.D.N., and A.I.S. wrote the manuscript.

## Competing interests

The authors declare no competing interests.
