## [Peer Review File · Nature Communications]

Structural mechanisms of TRPV6 inhibition by ruthenium red and econazoleREVIEWER COMMENTS

Reviewer #1 (Remarks to the Author):

Neuberger and colleagues report interesting structural pharmacology of the TRPV6 ion channel, which is implicated in epithelial calcium uptake. The cryoEM structure of TRPV6 in complex with ruthenium red revealed its binding in the selectivity filter. Thus, ruthenium red works like a bottle cork. In contrast, the antifungal drug econazole binds to a peripheral location and displaces a cholesterol lipid to inhibit the channel. Both inhibitors result in a closed channel pore. Structural findings have been further confirmed by mutagenesis studies. Overall, the cryoEM structures are of high quality, and the structural and functional studies have revealed new insights into TRPV6 inhibition. The manuscript is well written and it would be of interest to the ion channel field. A few points need to be addressed before publication.

Major:

Fig. 4 aims to show that econazole displaces cholesterol to cause inhibition. Fig. 4c and 4d illustrate the binding sites of econazole and CHS, respectively. It is still not clear what different modes of binding interactions lead to stabilization of the open state (CHS) vs allosteric inhibition (econazole) that results in closure of the bundle-crossing gate. Additional figures comparing detailed molecular interactions of econazole and CHS would be necessary to explain the distinct consequences at the same binding site. In addition, for mutations at this site reducing econazole inhibition, what are the effects on CHS stabilization? Do the mutations result in channels with reduced constitutive activity? To justify the conclusion that econazole displaces cholesterol to cause inhibition, it would also be important to determine whether CHS still binds to this site (maybe also weaker) by obtaining an additional structure of a representative mutation at this site (such as F472A or W495A).

Minor:

Fig. 3 a-c, showing EM densities for the pore region, including pore-facing residues, would be important for comparison of the ion pore properties under different conditions. There are non-protein densities in the central cavity of the econazole structure (Fig. 3c). How about the corresponding regions in the other structures?

Fig. 4 Panels were labelled a-e in the figure but described as a-c in the figure legend.

Reviewer #2 (Remarks to the Author):

TRPV6 plays an important role in Ca uptake by epithelial cells and has implications in diseases, including cancer. Therefore, TRPV6 inhibitors have clinical potential for the treatment of TRPV6-linked cancers. In this manuscript, Neuberger et al. investigated inhibitors of TRPV6 by combining cryo-EM, calcium imaging, and mutagenesis. They first solved the high-resolution structure of TRPV6 in open state in nanodisc, which is the highest open state structure of TRPV6. They further determined the structures of TRPV6 in complex with two inhibitors – the antifungal drug econazole and the universal ion channel blocker ruthenium red (RR). Thanks to the high resolution, they clearly defined the binding sites of the drugs, with econazole binding to an allosteric site at the periphery of the channel, replacing a cholesterol lipid; with RR binding to the middle of the ion channel selectivity filter. They compared the structures of the inhibited state and open state and elucidated an inhibitory mechanism whereby both inhibitors act similarly to close the gate despite binding to different sites. They also found that despite the high sequence similarity between TRPV5 and TRPV6, and despite the 100% conserved residues coordinating econazole in TRPV5 and TRPV6, the binding site for econazole is different from the previously reported site in TRPV5. In addition, the authors found that the location of RR binding to TRPV6 and the mechanism of RR inhibition differed from those in the CALHM channel and K2P channel.

Overall, this is an interesting and important study that will benefit drug development for the treatment of TRPV6-linked diseases, especially cancer. The manuscript is well written, the structural data is of

high quality and the structural analysis is thorough. Below are some minor comments on this already excellent manuscript.

1. Extended Data Fig. 1: the local resolution estimations do not seem to be consistent with the overall FSC-based resolution. For instance, the majority of the TRPV6_{eco} is in the range of 3-3.25Å, the TRPV6_{open}(GDN) is in the range of <2.5 Å, and the TRPV6_{open}(cNW11) and the TRPV6_{RR} are in the range of 2.75-3.25Å. However, the claimed resolutions of TRPV6_{open} (cNW11) and TRPV6_{RR} are higher than TRPV6_{open}(GDN). The authors should double-check the local resolution estimation.
2. The authors should provide an ED figure summarizing all available binding sites of TRPV6 and close homologues such as TRPV2 and TRPV5, which will be beneficial for the readers.
3. The authors defined the RR at the selectivity filter and econazole at the S4-S5 linker. It would be nice to include sequence alignment between TRPV channels highlighting key coordinating residues. In addition, there is no data from functional analysis using electrophysiology to validate the key coordinating residues. It would be nice to include the data.
4. The label in Figure 4 is not correct, please revise.

We are very thankful to all Reviewers for their excellent suggestions. We have made changes in the manuscript accordingly with the details outlined in our responses below.

Reviewer #1 (Remarks to the Author):

Neuberger and colleagues report interesting structural pharmacology of the TRPV6 ion channel, which is implicated in epithelial calcium uptake. The cryoEM structure of TRPV6 in complex with ruthenium red revealed its binding in the selectivity filter. Thus, ruthenium red works like a bottle cork. In contrast, the antifungal drug econazole binds to a peripheral location and displaces a cholesterol lipid to inhibit the channel. Both inhibitors result in a closed channel pore. Structural findings have been further confirmed by mutagenesis studies. Overall, the cryoEM structures are of high quality, and the structural and functional studies have revealed new insights into TRPV6 inhibition. The manuscript is well written and it would be of interest to the ion channel field. A few points need to be addressed before publication.

We are very thankful to Reviewer #1 for kind words about our work.

Major:

Fig.4 aims to show that econazole displaces cholesterol to cause inhibition. Fig. 4c and 4d illustrate the binding sites of econazole and CHS, respectively. It is still not clear what different modes of binding interactions lead to stabilization of the open state (CHS) vs allosteric inhibition (econazole) that results in closure of the bundle-crossing gate. Additional figures comparing detailed molecular interactions of econazole and CHS would be necessary to explain the distinct consequences at the same binding site. In addition, for mutations at this site reducing econazole inhibition, what are the effects on CHS stabilization? Do the mutations result in channels with reduced constitutive activity? To justify the conclusion that econazole displaces cholesterol to cause inhibition, it would also be important to determine whether CHS still binds to this site (maybe also weaker) by obtaining an additional structure of a representative mutation at this site (such as F472A or W495A).

To clarify molecular details of how binding of econazole can stabilize the closed state, we made a superposition of the apo and econazole-bound structures and added panel f to Figure 4 that shows a close-up view of this superposition focusing at the econazole binding site. This superposition clearly illustrates that binding of econazole between F472 and W495 creates a void that allows Q473 and M474 side chains to move closer to F472 and W495 and correspondingly away from R589. Separation of Q473 and R589 results in breakage of the salt bridge connecting them in the open state (McGoldrick *et al.*, 2018). The loss of this salt bridge allows the lower portions of S6 helices to rotate by $\sim 100^\circ$, leading to breakage of D489 and T581 hydrogen bond that also stabilizes the open state (McGoldrick *et al.*, 2018). The rotated lower portions of S6 expose the side chains of L574 and M578 towards the pore center and hydrophobically seal the pore to prevent ion conductance, thus converting the channel into the closed, non-conducting state.

Based on this better clarified view on the molecular basis of econazole inhibition, we do not think that mutations cause CHS stabilization. Instead, they weaken econazole binding. Lipids

simply have a better chance to occupy the same site whenever econazole is not present there. However, the mutations do not have to change the affinity to lipids.

We have also taken a more careful look at the density of lipid in econazole binding site in the apo state. In the best-resolution structure of the apo-state TRPV6, this density does not resemble the shape of CHS enough to justify the fit of CHS. In fact, in other better resolution structures of hTRPV6, this density looks more like an acyl chain than CHS. We have previously modeled this density as a tail of phosphatidylcholine (Bhardwaj *et al.*, Sci Adv 2020). The distinction becomes obvious when comparing this density to the density of the three prominent CHS binding sites, where the shape of CHS is undoubtedly clear. Therefore, in our econazole-free structures, including apo-state and RR-bound structures, we replaced the CHS molecule at site 4 (econazole binding site) with an acyl chain of phosphatidylcholine, as in our previous best-resolution structures (Bhardwaj *et al.*, Sci Adv 2020).

We also noticed that lipid density at site 4 is relatively weak and appears only in the highest-resolution TRPV6 structures. The quality of lipid densities in cryo-EM maps fluctuates from collection to collection. It may depend on the nature of lipid, site specificity, heterogeneity of the surrounding lipids, protein flexibility, microscope alignment, overall sample, collection and data quality, number of particles, processing strategy etc. Given that the interpretation of either outcome of such structural experiment is going to be ambiguous and may be misleading, we prefer not solving structures of F472 and W495 mutants. Instead, we made our interpretation of the mechanism of econazole inhibition more cautious and removed mentions of CHS at this site accordingly (page 7, second paragraph). We have also corrected the text to better explain our interpretation of the econazole inhibitory mechanism (page 10, last paragraph).

Minor:

Fig. 3 a-c, showing EM densities for the pore region, including pore-facing residues, would be important for comparison of the ion pore properties under different conditions. There are non-protein densities in the central cavity of the econazole structure (Fig. 3c). How about the corresponding regions in the other structures?

When reducing the map contour level to the level of noise, there are very small and weak blobs of density in the pores of other structures, which can be interpreted as water molecules, but nothing comparable to large density in the central cavity of the econazole-bound structure. We have now mentioned this in the text (page 7, first paragraph).

Fig. 4 Panels were labelled a-e in the figure but described as a-c in the figure legend.

Thank you for noticing. We have fixed the panel labeling.

Reviewer #2 (Remarks to the Author):

TRPV6 plays an important role in Ca uptake by epithelial cells and has implications in diseases, including cancer. Therefore, TRPV6 inhibitors have clinical potential for the treatment of TRPV6-linked cancers. In this manuscript, Neuberger *et al.* investigated inhibitors of TRPV6 by combining cryo-EM, calcium imaging, and mutagenesis. They first solved the high-resolution

structure of TRPV6 in open state in nanodisc, which is the highest open state structure of TRPV6. They further determined the structures of TRPV6 in complex with two inhibitors – the antifungal drug econazole and the universal ion channel blocker ruthenium red (RR). Thanks to the high resolution, they clearly defined the binding sites of the drugs, with econazole binding to an allosteric site at the periphery of the channel, replacing a cholesterol lipid; with RR binding to the middle of the ion channel selectivity filter. They compared the structures of the inhibited state and open state and elucidated an inhibitory mechanism whereby both inhibitors act similarly to close the gate despite binding to different sites. They also found that despite the high sequence similarity between TRPV5 and TRPV6, and despite the 100% conserved residues coordinating econazole in TRPV5 and TRPV6, the binding site for econazole is different from the previously reported site in TRPV5. In addition, the authors found that the location of RR binding to TRPV6 and the mechanism of RR inhibition differed from those in the CALHM channel and K2P channel.

Overall, this is an interesting and important study that will benefit drug development for the treatment of TRPV6-linked diseases, especially cancer. The manuscript is well written, the structural data is of high quality and the structural analysis is thorough. Below are some minor comments on this already excellent manuscript.

We are grateful to Reviewer #2 for the generous opinion about our work.

1. Supplementary Fig. 1: the local resolution estimations do not seem to be consistent with the overall FSC-based resolution. For instance, the majority of the TRPV6eco is in the range of 3-3.25Å, the TRPV6open(GDN) is in the range of <2.5 Å, and the TRPV6open(cNW11) and the TRPV6RR are in the range of 2.75-3.25Å. However, the claimed resolutions of TRPV6open(cNW11) and TRPV6RR are higher than TRPV6open(GDN). The authors should double-check the local resolution estimation.

Thank you for bringing up the inconsistencies. The difference originates from the fact that cryoSPARC and Relion software presumably have different algorithms for the local resolution estimation. We have corrected the local resolution estimates by using exclusively Relion software for calculating Resmap.

2. The authors should provide an ED figure summarizing all available binding sites of TRPV6 and close homologues such as TRPV2 and TRPV5, which will be beneficial for the readers.

As requested, we have added this figure, which became a new Supplementary Figure 4.

3. The authors defined the RR at the selectivity filter and econazole at the S4-S5 linker. It would be nice to include sequence alignment between TRPV channels highlighting key coordinating residues. In addition, there is no data from functional analysis using electrophysiology to validate the key coordinating residues. It would be nice to include the data.

We highlighted the key residues involved in binding of RR and econazole in the new Supplementary Figure 5 sequence alignment.

Comparative analysis of TRPV6 mutants using electrophysiology is always problematic as it requires measurements of concentration-dependence of TRPV6 current inhibition. Because of constitutive activity of TRPV6, it is always difficult to separate the leak and TRPV6-mediated currents in the whole-cell patch-clamp recordings, the problem becoming more significant when aiming to reliably measure changes in IC₅₀ of an inhibitor. For this reason, we use Fura-2-based measurements, which report specific calcium influx through the TRPV6 pore. Functional characterization of RR is additionally complicated by the fact that it is a dye. The absorption spectrum for Ruthenium red has a peak at 534 nm, which is close to the emission wavelength of 510 nm used in Fura-2-based experiments. We have previously tried using RR in our whole-cell patch-clamp recordings and ended up replacing the components of the rig after not being able to wash RR off.

Mutagenesis experiments with RR would also be very tricky for interpretation. Because the RR binding site is the selectivity filter and RR is mostly coordinated by backbone carbonyl oxygens, binding site mutations will inevitably alter all channel functions, not just RR block. According to LigPlot software (Laskowski and Swindells, 2011) and visual inspection of TRPV6_{RR}, the side chains of D542 and T539 may also contribute to RR binding. However, mutations of these residues were previously shown to alter TRPV6 function as well. Indeed, alanine mutation of D542 significantly altered TRPV6 ion permeation (Nilius *et al.*, JBC 2001). Correspondingly, by doing electrophysiological recordings from RR site mutants, it will be hard to separate changes in ion channel function from changes in RR inhibition. So we decided not to include such experiments into our manuscript to avoid misinterpretation of data.

Luckily, cryo-EM density for RR in the high-resolution TRPV6_{RR} structure is so evident visually that it leaves no doubts about the identity and location of the RR binding site in TRPV6.

4. The label in Figure 4 is not correct, please revise.

Thank you for noticing. We have fixed the panel labeling.

REVIEWER COMMENTS

Reviewer #1 (Remarks to the Author):

I thank the authors for their revisions and am happy with this very nice manuscript.